# Simulating the Changes of Invasive *Phragmites australis* in a Pristine Wetland Complex with a Grey System Coupled System Dynamic Model: A Remote Sensing Practice

Danlin Yu [1],*⊙, Nicholas A. Procopio [2]⊙ and Chuanglin Fang [3]

[1] Department of Earth and Environmental Studies, Montclair State University, Montclair, NJ 07043, USA
[2] Division of Science and Research, New Jersey Department of Environmental Protection, Trenton, NJ 08625, USA
[3] Center for Urban and Regional Planning Design and Research, Institute of Geographic Science and Natural Resources Research, Chinese Academy of Sciences, Beijing 100045, China
* Correspondence: yud@mail.montclair.edu; Tel.: +1-973-655-4313

**Abstract:** Biological invasion has been one of the reasons that coastal wetlands gradually lose their ecological services. The current study investigates the spread of a commonly found invasive species in coastal wetlands in Northeastern US, the *Phragmites australis*. Within a relatively pristine wetland complex in coastal New Jersey, we collected high-resolution multispectral remote sensing images for eight years (2011–2018), in both winter and summer seasons. The land cover/land use status in this wetland complex is relatively simple, contains only five identifiable vegetation covers and water. Applying high accuracy machine learning algorithms, we are able to classify the land use/land cover in the complex and use the classified images as the basis for the grey system coupled system dynamics simulative model. The simulative model produces land use land cover change in the wetland complex for the next 25 years. Results suggest that *Phragmites australis* will increase in coverage in the future, despite the stable intensity of anthropogenic activities. The wetland complex could lose its essential ecological services to serve as an exchange spot for nekton species from the sea.

**Keywords:** biological invasion; *Phragmites australis*; grey system; system dynamics simulative model; machine learning; supervised classification

## 1. Introduction

Invasive species have been regarded as a parasite problem for many local ecosystems across the world [1]. The invasiveness of such species often poses a significant threat to native species and the ecosystem they invaded because they take over resources and encounter less competition from the native species or lack of natural predators, and often adapt well to their invaded ecosystems, and could harm indigenous taxa through predation, habitat modification, cross-species hybridization and alteration of ecosystem processes [2]. It is commonly agreed and modeled that biological invasions are the primary reasons for substantial biodiversity declines in any native ecosystems [1]. Biological invasions also cause very high economic losses to society and monetary expenditures associated with the management of these invasions [3]. Biological invasion is a particularly troublesome problem for many of the ecologically protected areas that are usually preserved from intensive human interventions either because of their ecological fragility or natural historical values. It was estimated that within these protected areas worldwide, invasive species cause a reported cost of USD 22.24 billion between 1975 and 2020 [4]. For the entire economy, the cost of biological invasion is almost unbearably high. One study finds through the synthesized economic impact data from the InvaCost database [5] that biological invasions have cost the North American economy at least USD 1.26 trillion between 1960 and 2017 [5]. The study further finds that the cost averages USD 2 billion per year in the early 1960s to

over USD 26 billion per year in the 2010s. The United States had the highest recorded costs. Another study that attempts a nationwide estimate of the economic costs to the United States of nonindigenous species concluded that annual costs exceed USD 120 billion or about USD 1100 per household annually [6]. The cost is primarily born in the agricultural and forestry sections. From 1960–2017, invasive species cost the agriculture and forestry sectors of the US an estimated USD 527.07 billion and USD 34.93 billion, respectively. Admittedly, the estimated economic cost in the US varies depending on studies, database extracted, and even models applied, the consensus is that it is unnecessarily high and soaring as anthropogenic activities intensify [7].

While biological invasion is a multidimensional issue that involves a host of species, both plants and animals, our current study focuses on one particular invasive species that are often found in tidal salty wetlands in the Northeastern US, the *Phragmites australis*. Although evidence suggests that a subspecies of *Phragmites australis* existed in North America prior to the European colonization, it is commonly believed that current populations represent an exotic/invasive subspecies of *Phragmites australis* [8–10]. This wetland plant species is present on every continent except Antarctica. Both native and non-native subspecies are found and thrive in the US, but usually, the non-native species quickly displace native wetland plants because of their relatively better environmental adaptation [11]. The introduction and spread of *Phragmites australis* took root in the tidal salt marsh wetlands due to built dikes in the early 20th century [9,12,13]. The dikes allowed the *Phragmites australis* to become a dominant species in these tidal marshes in the Northeastern US. The *Phragmites australis* can grow at a height of greater than 4.6 m. The species is usually the tallest grass species in wetland, estuary, and marsh ecosystems, and can alter the ecology of these wetland systems making them less suitable as habitats for many species of flora and fauna [11]. For instance, *P. australis* has been found to elevate the marsh planform and smooth out the marsh surface by filling in the microtopography [14–16]. The elevated marsh planform alters the hydroperiod of the tidal marsh, preventing regular daily seawater exchange with the marsh wetlands which many near sea nekton species obtain their food from [15]. Similarly, the filled microtopography changes the access to the marsh plain by resident and transient organisms [9,14,15,17–20], which could significantly impact local ecosystems and cause the wetlands to recede and even disappear. Moreover, the steep banks of *P. australis*-lined tidal creeks may also negatively influences the survival of small fishes by exposing them to increased predation within the marsh wetlands [14,15,21].

On the other hand, tidal wetlands are some of the most productive ecosystems on Earth and critical habitats for many species. According to the National Oceanic and Atmospheric Administration (NOAA) Fisheries' coastal wetland habitat report [22], there are at least six benefits of the coastal wetlands, including sustainable fisheries (which in 2018, U.S. commercial and recreational fisheries supported 1.7 million jobs and contributed USD 238 billion in sales), tourism and recreation (more than a third of all U.S. adults have utilized the natural wetlands for some types of tourism and recreation activities. In 2018, anglers along in the coastal wetlands generated more than USD 72 billion in sales and supported 470,000 U.S. jobs), energy production, clean water (wetlands is a natural water purifier, the native vegetation of the wetlands is able to filter sediment and absorb pollution carried by water), flood and storm protection (vegetation coverage in the wetlands can lower overall flood heights and drastically reduce the speed of wind, protecting people, property, infrastructure, and agriculture from devastating flood and hurricane damages. It is estimated that this protection saves coastal communities USD 23 billion each year in the US), and coastal blue carbon (this is because salt marshes, seagrass beds, and mangroves can remove greenhouse gases such as carbon dioxide from the atmosphere and store them in plants and in the soil) [22]. The increased biological invasions, however, are rapidly deteriorating the benefits of the coastal wetlands by altering the microtopography of the wetlands, replacing native species, and changing the drainage properties of the wetlands.

The scientific and policy communities have long concluded that anthropogenic activities are among the most detrimental causes for coastal communities and ecosystems

that rely on delicate and fragile exchanges between the near sea and the tidal marsh wetlands [23–29]. Anthropogenic activities are also the primary cause of invasive species in many local communities creating a direct threat to the persistence and maintenance of healthy wetlands [23,24,28]. While changes in ecosystems are anticipated and sometimes necessary for their evolution, rapid changes have unanticipated consequences that require close monitoring and evaluation to support a sustainable system without catastrophic disruptions. Protecting, conserving, and restoring coastal wetlands is not only necessary to preserve a piece of nature and promote harmonious coexistence between human beings and the coastal environment, but also a critical part of the enrichment of human lives as well.

Protecting, conserving, and restoring the coastal wetlands requires an accurate monitoring, evaluating, and simulative system of coastal wetlands, especially when the wetlands are threatened by the existence and potential expansion of invasive species under the rapid but often latent influence of anthropogenic activities. In such cases, knowing the distribution of various native and invasive species within coastal wetlands is often the first step in many restoration efforts [9,13,18,19,25,29,30]. Mapping through GIS and mapping and monitoring through remote sensing images and technologies are the most commonly employed strategies in achieving such tasks. For instance, Allen and colleagues [13] used multi-date and multi-sensor remote sensing images to map coastal wetlands, focusing on the impact of the invasive *Phragmites australis* in swamp forests and pocosins, marshes, and shrub–scrub environments. By combining spaceborne multi-date synthetic aperture radar (SAR) images and airborne light detection and ranging (LiDAR) elevation (bare earth elevation and vegetation height) images, they conclude that the combination has great potential to monitor wetland change, sea level rise, and invasive species. In another study [31], Ghioca-Robrecht and colleagues used QuickBird multispectral satellite (DigitalGlobe, Westminster, CO, USA) images taken in September 2002 and April 2003 to map emergent wetland vegetation communities within a diked wetland at the western end of Lake Eric. They found that the Multiseason QuickBird imagery is promising for distinguishing certain wetland plant species (*P. australis*), though cautioned that care needs to be taken when monitoring highly cultivated land since anthropogenic activities could alter the landscape rather quickly. In another study [12], Nielsen and colleagues produced a 30-m resolution wetland change probability map for the U.S. mid-Atlantic region by applying an outlier detection technique to a base classification provided by the National Wetlands Inventory (NWI). Their approach found that the change probabilities identify areas for closer inspection of change cause, impact, and mitigation potential. A more recent study utilizes airborne unmanned vehicles [11] to map the *P. australis* along the US Pearl River delta in southeastern Louisiana. The study applied a wavelet-based texture analysis to map the extent of *Phragmites australis* that was reasonably accurate with an average accuracy of 85% and an average kappa accuracy of 70%.

These studies, among many others, suggest that monitoring and evaluating biological invasions (*Phragmites australis* as in our current study) with remote sensing and GIS technologies have been gradually established as a routine practice. The constant, year-around, and automatic recording of the Earth observatory platforms through satellites carried sensors and the on-demand, object-oriented, highly customizable drone-acquirable images provide two vast information sources for tackling the task of accurately monitoring and evaluating the degrees of biological invasions, which often provide invaluable information for invasive species management and restoration of native ecosystems. On the other hand, while monitoring and evaluating biological invasion through remote sensing and GIS produces valuable information, seldom were there studies that attempt to predict or simulate how the ecosystem might evolve in the future. Understandably, prediction often requires a relatively thorough understanding of the structure, the interaction, the participants/components, and many other factors within a system needed to predict its evolution. For the coastal wetland system that is also under the influence of usually unobservable and latent anthropogenic activities, a "thorough" understanding is not easily attenable. Yet on the other hand, while the structures of an ecosystem could remain unrevealed, with

accumulated temporal data on identifiable land use land cover components through the satellite images, we could attempt to establish covariations among the land use land cover components of the ecosystem to simulate the evolutionary trajectory of the ecosystem. The invasiveness of the *P. australis* is reflected in the satellite images as a land cover type. Although anthropogenic activities are usually not directly observable through satellite images, the influence will be reflected in the interactions between different land cover types—which can be captured through modeling and simulating using mined covariations among them [32–35]. This particular area of investigation, however, has been virtually unaddressed in the vast number of wetland assessment and management studies and programs developed to date.

In this regard, this study focuses on two key tasks for coastal wetland invasiveness management purposes. We select a small area in the lower Wading River Complex (WRC) in New Jersey (USA) as our test site (Figure 1). First, we intend to apply available machine learning algorithms to a series of high-resolution (0.5 m spatial resolution) satellite images obtained from 2011–2018 with a pre-created training site layer (Figure 1), during both summer and winter, to identify the extent of the invasiveness of *P. australis*. Second, through exploratory analysis of the covariations among different land cover types within the WRC, we aim to establish a system dynamic model [32,33,35–37] among the different land cover types. Our purpose is to answer the question of how individual components of the coastal wetland are *inter-connected* to form a functional whole, and how the *Phragmites australis* will likely expand in the near to medium-term future. The information, we hope, could provide scientific evidence and support for active policymaking to curb further biological invasions of pristine coastal wetlands.

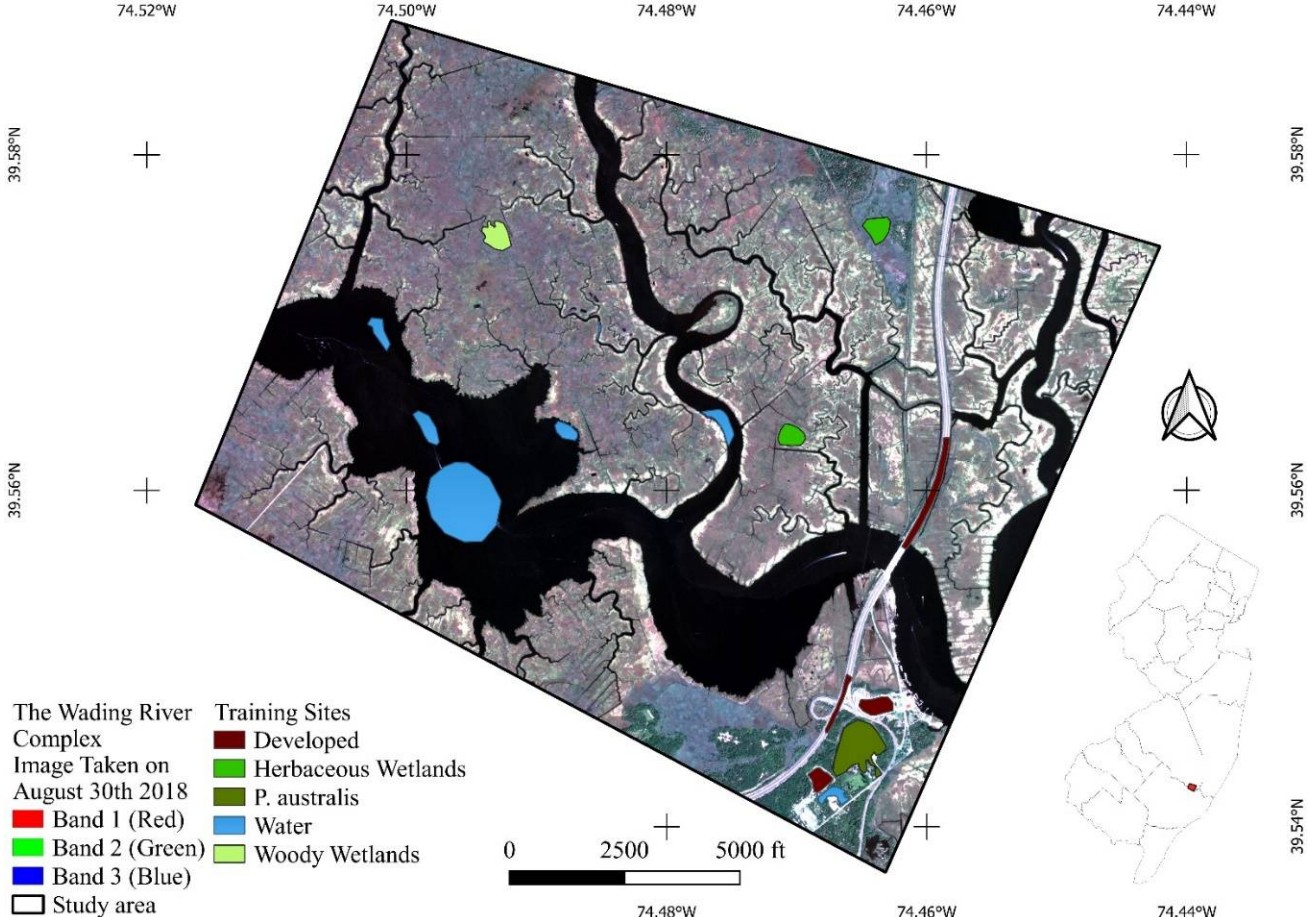

**Figure 1.** Regional location within the WRC where satellite imagery was evaluated.

The study is organized as follows. Following this introductory section, we will briefly introduce the study area, the data, and the methodology. This is followed by the analytical results and discussion. The study concludes with a summary of the findings.

## 2. Materials and Methods

### 2.1. Study Area

The study is conducted in a relatively pristine coastal wetland complex in the lower Wading River in the state of New Jersey, USA (Figure 1). This study area is a minimally disturbed watershed located in the area between the mouth of the Mullica River and the headwaters of the Wading River and its tributaries, we refer to this area as the Wading River Complex (WRC). The entirety of the WRC has a total area of 590 km$^2$. The areas that we obtain data for final analysis, however, only contain 20 km$^2$ of coastal tidal wetlands (Figure 1). Here, land use is minimally impacted by humans, and the watershed is dominated by coastal wetlands with herbaceous species and water, followed by woody wetlands, developed lands (building and roads), and *Phragmites australis*.

### 2.2. Using High-Resolution Multispectral Images to Characterize Land Covers

High-resolution multiband satellite images for the 20 km$^2$ area of the Wading River Complex were purchased from WorldView 2 (https://www.usgs.gov/centers/eros/science/usgs-eros-archive-commercial-satellites-cdp-imagery-worldview-2, accessed on 20 June 2019) and Pleiades (https://eos.com/pleiades-1/, accessed on 20 June 2019) satellites for summer and winter times from 2011–2018 (2012 summer multiband image is unavailable). Images contain the infrared (wavelength 750–950 nm), red (wavelength 600–720 nm), green (wavelength 490–610 nm), and blue (wavelength 430–550 nm) bands. Images from both sources have a 50-cm spatial resolution. All the images were geo-rectified and haze-removed prior to being analyzed. The images were projected with a UTM 18N coordinate system and WGS 84 datum. The images were not re-projected to the New Jersey State Plane coordinate system to ensure accurate digitization and ensuing data extraction as well as to avoid potential information loss during the resampling and assessment processes (Figure 1).

Prior to analyzing the land use land cover patterns, detecting its change via classification, and exploring the covariations among different land covers in the study area, we examined the river system within the study area to determine whether there are significant changes in the pattern, distribution, and complexity of the watercourse channels. This practice is to determine if the morphological change of the river channels might also contribute to the change of *Phragmite australis* coverage in the wetland complex. This result will be used to determine the simulative model frame in later analysis. For this task, the WRC study area was digitized from the satellite imagery in two formats: one in a polygon, and the other in polyline format using QGIS software (3.12, https://www.qgis.org/en/site/, accessed on 20 June 2019). The polygon layers were created and then converted to polyline layers in the study area from an approximate vector perspective. These layers were carefully digitized for winter and summer periods for all years (2011–2018). The purpose of this practice is to detect any morphological changes in the river channels during the study period.

To characterize land use land cover using the remote sensing images, we adopted machine-learning-supported supervised image classification strategies. An extensive review of the literature [18,38–52] suggests that the random forest, Support Vector Machine, and Convolutional Neural Network algorithms are among the most commonly applied approaches that can provide satisfactory supervised classification accuracy. Detailed technological discussions of the three algorithms are abundant in the literature, for instance, see [53–55], hence will not be repeated here.

Based on image inspection and land use classification from the 2016 National Land Cover Database (NLCD) [56], five primary land cover classes in the study area were identified. The land cover codes, and land cover types are: 11-Water, 20-Developed (contains 21–24 subclasses of the NLCD), 40-*Phragmites australis* (contains 41–43 subclasses

of the NLCD, the invasive species in this wetland), 90-Woody Wetlands, and 95-Emergent Herbaceous Wetlands. A training data set (polygon layer) (Figure 1) for supervised image classification was generated utilizing the Semi-automatic Classification Plug-in tool for QGIS 3.12 (https://www.qgis.org/, accessed on 20 June 2019) via digitizing the typical areas of the five land cover types on each image.

To compare the accuracy of the different algorithms, oftentimes we report the overall accuracy (the correctly classified pixels divided by the total number of pixels in an image) and the Cohen's Kappa coefficient [49]. Cohen's Kappa coefficient (KC) takes the form:

$$\text{KC} = \frac{CD - \sum_i^n (TC_i * TR_i)}{1 - \sum_i^n (TC_i * TR_i)} \tag{1}$$

where *CD* is the correctly classified pixels of the classification; $TC_i$ is the total number of pixels that belong to the *i*th category of land cover, $TR_i$ is the total number of pixels that are classified as the *i*th category of land cover. In essence, KC measures how well the classification does compare to just randomly assigning values to each category during the classification. KC values suggest how well the classification is away from just randomly assigning values. Usually, an 80% or above KC suggests a reasonably accurate classification.

### 2.3. The Grey System Supported System Dynamic Simulative Model

One of the goals of scientific research is to predict or forecast the future based on available information. Even though the land use land cover system in the WRC is relatively simple since there are a limited number of land cover types and the water area and river system complexity remain stable, prediction of how land use land cover changes is much more complex than predicting the trend of one or two key land use land covers (such as the *P. australis*) of the system because of two reasons. First, land use land cover changes within the wetland complex are caused by an array of factors, including human activities, climatic conditions, and changes in other species in the wetlands, among many others. These factors, however, usually are not directly observable or recorded in any existing database. Second, influences on different land use land cover types are often temporally and spatially confounding. Even when data are available, trying to separate contributing factors tends to be futile. On the other hand, these confounding contributions from the latent factors also provide observable covariation patterns among the different land use/land cover types. These covariation patterns, while not indicative of cause-effect relationships, tend to remain stable within a stable ecosystem, and do not change over short to medium terms. Based on studies of similar nature [35,57–62] and the stable covariation patterns, we found the system dynamic (SD) simulative model might serve the purpose of simulating how different land use land cover types change in the future.

SD modeling has a wide range of applications since it was first promoted by Forrester and colleagues [63,64] in the early 1960s [34,65–79]. When the modeling scheme was promoted, it was intended to solve a potential sustainable development problem—to predict how the world would progress under the then conditions (socioeconomic, environmental, industrial development, and the like). The book, *The Limits to Growth* [80], is a direct application of the SD model in this regard, and the results are still widely discussed today [81–84]. The major task of this study was to establish a system dynamic model that can simulate the dynamic changes of the land cover in the Wading River Complex. To build an effective system dynamic model, it is necessary to identify the different components in the system. In this pristine wetland complex, because of the limited human activities, and minimally disturbed biogeographical conditions, the identifiable and collectible data through remote sensing images is simple. It only includes the five different types of land cover that were identified and classified through remote sensing images.

The appealing characteristic of the SD is that it is based on a set of indicators that represent the components of a particular system (land use land cover in our relatively simple system) and takes into consideration the feedback-based interactions among individual indicators explicitly [58–62]. These feedback-based interactions are usually derived from

exploring the covariation patterns among the observable system components. Although it might be an overstatement that the SD model can "predict" the status of the wetland land cover of the WRC in the future, the results generated from the SD model will provide the basis for informed decision-making and potentially reduce possible pitfalls such as insufficient evidence for causal relationships due to the lack of information that can be used for predictive analysis. System dynamic simulative models are sometimes criticized for their potentially unreasonable definition of the relationships between various components of the system. This is because the relationships are established based on explored covariation patterns. In the wake of the recent boom of big data analytics, it was found that the covariations among different system components instead of confirmatory cause–effect relationships might serve to understand a complex system much better. This is because, for a complex system, the interactions among the system components are usually highly convoluted and can hardly be described with clear pathways between causes and effects (or even which are causes and which are effects). On the other hand, while the latent causes of the change for one component of the system might not be easily identifiable, changes in the observable system components can be compared, and careful mining of such observations could produce meaningful and useful covariations that enable us to simulate the trend of the system without knowing exactly the cause–effect relationships among system components. This particular philosophy is shared with both the system dynamic simulative models and the current big data analytics.

In a typical system dynamic model, depending on the individual indicators and their interactions, we can roughly divide them into four primary groups: the stock, the process, the auxiliary variables, and the flow. The stock variable stores the basic (and core) states of the system. It is governed by in- and out-rates that are often impacted by the process and/or auxiliary variables. The processes (or ongoing activity) in the system determine the contents of the stock. Auxiliary variables are system variables that often dictate the rates at which processes operate. Flow (or inter-relationship) represents the intricate connections among all components of a system [85]. The relationships among the stock, the process the auxiliary variables and the flow can be collectively expressed as:

$$
\begin{aligned}
S_{i(t+1)} &= S_{i(t)} + R_{i(t)} dt \ (i \in [1, \cdots, n]) \\
R_{i(t)} &= f\left(A_{j(t)}, S_{k(t)}, t\right) \ (j, k \in [indicators \ related \ with \ i_{th} \ stock \ changing \ rate] \\
A_{j(t)} &= g\left(A_{l(t)}, S_{k(t)}, t\right) \ (l, k \in [indicators \ related \ with \ j_{th} \ auxilliary \ indicator]
\end{aligned}
\tag{2}
$$

where $S_{i(t)}$ is the value of the *i*th stock indicator at time *t*, $R_{i(t)}$ is the changing rate of the *i*th stock indicator at time *t*, *n* is the number of stock indicators, $A_{j(t)}$ ($A_{l(t)}$) is the *j*(*l*)th auxiliary indicator that is related with the *i*th stock indicator's changing rate or the *j*th auxiliary indicator at time *t*. *f* and *g* are functions derived from data analysis. It can be seen from these equations that all the variables depend on time. Oftentimes, in practice, the only difference between a stock and an auxiliary variable is whether the variable is determined by its rate or directly by other auxiliary and/or stock variables. Auxiliary variables and rates can be derived following similar manners. The interactions and interconnections among the variables are explicitly linked through a networked feedback loop. For illustration purposes, a simplified networked feedback loop for the land use land cover types in the WRC can be illustrated in Figure 2.

### 2.4. A Grey System Modification

As expressed in Equation (1), in a typical system dynamic simulative model, time is explicitly expressed in the equations as the incremental factor in the stock variable or an embedded determinant in the auxiliary and rate functions. However, if relationships derived via statistical exploration contain too much uncertainty and/or introduce too much rigidity, which could cause a long-term prediction to be unreliable [69,86]. Time can also

be explicitly used as a determinant. Incorporating time explicitly in the relationships is conducted with a grey system time series simulation procedure that we detail below.

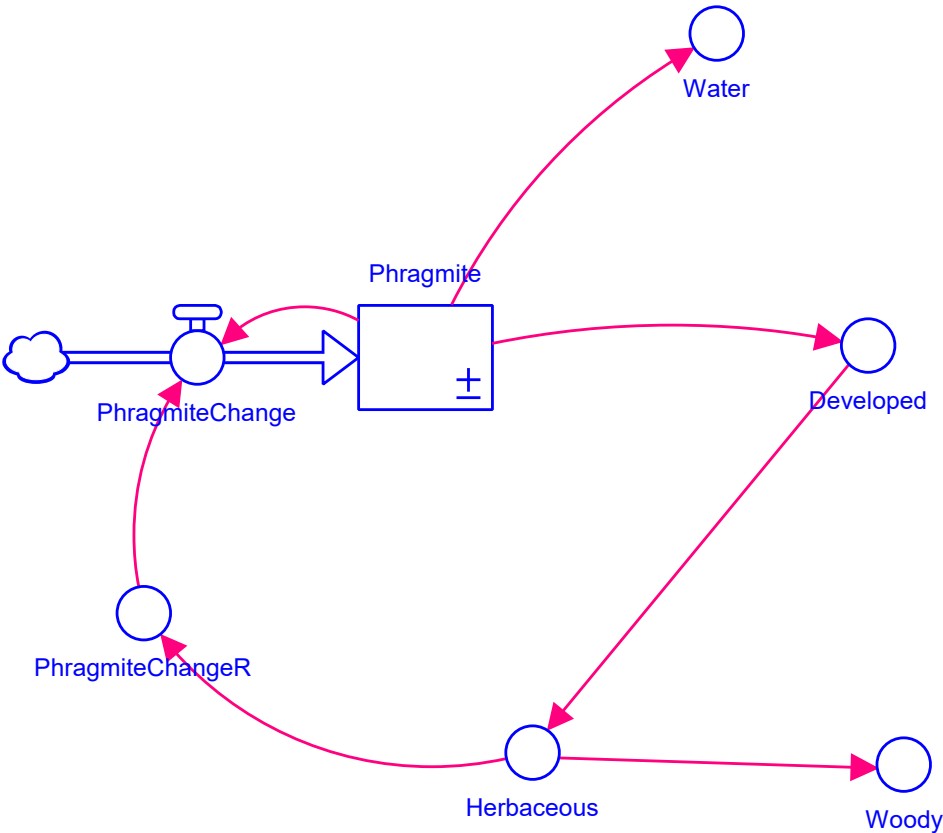

**Figure 2.** The conceptual system dynamic model structure of the five land use types in the Wading River Complex.

The theory of the grey system was introduced during the 1980s by Deng [87]. After its introduction, the theory soon was regarded as an attractive way to understand, model, and incorporate uncertainty that is inevitable in data analysis, especially when recorded data are sparse [88]. As a matter of fact, in the software package developed by Liu and associates [89], 4 data points at a minimum can be used to generate a tentative model to investigate the data series' inherent uncertainty and produce a simulation that fits relatively well with the original data. The central idea of a grey system simulation model is to find the inherent regularity within the data that is represented by a seemingly irregular data series or a data series that is too short to exhibit regularity. To do so, the target data series are often re-generated via accumulative addition or deduction operations [89].

Suppose we have a data series $X^{(0)} = (x^{(0)}(1), x^{(0)}(2), \ldots, x^{(0)}(n))$, a first order accumulative addition operation on this series will produce the first order accumulatively added (1-AA) series: $X^{(1)} = (x^{(1)}(1), x^{(1)}(2), \ldots, x^{(1)}(n))$, where $x^{(1)}(k) = \sum_{i=1}^{k} x^{(0)}(i)$. On the other hand, a first order accumulative deduction operation will produce the first order accumulatively deducted (1-AD) series: $X^{(1)\mathrm{d}} = (x^{(1)\mathrm{d}}(1), x^{(1)\mathrm{d}}(2), \ldots, x^{(1)\mathrm{d}}(n))$, where $x^{(1)\mathrm{d}}(k) = x^{(0)}(k) - {}^{(0)}(k-1)$. Often the data series produced via accumulative addition or deduction will exhibit stronger regularity than the original one. Except for accumulative addition and deduction, another important operation, the mean operation with consecutive neighbors, is often employed and critical in grey model simulation and prediction. The operation will generate a new data series: $Z = (z(2), z(3), \ldots, z(n))$, where $z(k) = 0.5\,x(k) - 0.5\,x(k-1)$, $z(1)$ is often omitted since there is no left neighbor for $x(1)$. The primary model that is developed for data series simulation and prediction is called the GM $(r, h)$ model, with $r$ the order of the accumulative operation (often accumulative addition), and $h$ the number of co-varying data series plus the target series. As pointed

out by Deng [90], GM (*r*, *h*) models are often a good fit for data simulation, for data series prediction, however, GM (1, 1) model is by far the most appropriate.

In our study, our primary purpose for introducing GM models into the system dynamic modeling scheme is to mitigate the potential rigidity and uncertainty that were generated via statistical analysis with relatively limited data points. For this purpose, and the argument by Deng [90], we focus on the GM (1, 1) model. The GM (1, 1) model is based on the target series, its 1-AA series, and the mean operation with consecutive neighbors generated from the 1-AA series. The basic form of the GM (1, 1) model is expressed as [89,90]:

$$x^{(0)}(k) + az^{(1)}(k) = b \quad (k = 2, \dots, n) \tag{3}$$

where *n* is the number of data points. *a* is regarded as the development coefficient, and *b* is the grey action measure. The development coefficient is a measure of the target series' changing trend and the inherent uncertainty, while the grey action measure reflects the internal relationships among data points [89]. From Equation (2), *a* and *b* can be derived via least squares operation [89]:

$$\begin{cases} a = \dfrac{\frac{1}{n-1}\sum_{k=2}^{n} x^{(0)}(k)\sum_{k=2}^{n} z^{(1)}(k) - \sum_{k=2}^{n} x^{(0)}(k)z^{(1)}(k)}{\sum_{k=2}^{n} \left(z^{(1)}(k)\right)^2 - \frac{1}{n-1}\left(\sum_{k=2}^{n} z^{(1)}\right)^2} \\ b = \frac{1}{n-1}\left[\sum_{k=2}^{n} x^{(0)}(k) + a\sum_{k=2}^{n} z^{(1)}(k)\right] \end{cases}. \tag{4}$$

With *a* and *b* obtained, the series can be simulated and forecasted as [89]:

$$x^{(0)}(k) = \left(\frac{b - 0.5a}{b + 0.5a}\right)^{k-2}\left(\frac{b - ax^{(0)}(1)}{b + 0.5a}\right) \tag{5}$$

Since *a* measures the inherent uncertainty of the target data series, from experimental analyses, Liu and colleagues conclude that a GM (1, 1) model with an *a* value that is within the range of [−0.3, 0.3] can be used for relatively longer term (up to 10–15) forecast [89,91]. A grey system time series simulation is often used as a supplementary choice to control and mitigate the rigidity introduced in regular system dynamic models [69,86].

By combining the conventionally obtained covariation relationship with the Grey System Series Simulation approach, we could then build the fully functional system dynamic simulative model that functions to simulate the covariation among the five different land use types. The general system dynamic model structure is presented in Figure 2. The model structure is simple since the dynamic is built entirely on the observed covariation among the five different land cover types. As aforementioned, all the explorations among the variables pertaining to the "covariation" among them. It is recognized that there are many other agents that are at work to drive the dynamics of the system. By observing the covariation, however, the system dynamic simulation might be able to capture intuitively the latent relationships that drive the dynamics of the system. Figure 2 is a simple manifestation of how such dynamics might be presented in a feedback loop structure. Simply put, one land cover type is regarded as co-varying with another land cover type. While the land cover type of *Phragmites australis* is our focus type, we use that land cover type as the hub to connect all other land cover types in the simultaneous dynamics.

## 3. Results and Discussion

### 3.1. Digitization Analysis

We have conducted visual inspection and map overlay analyses of the digitized polygons and polylines layers over the eight years and 15 images. Map overlay was performed in QGIS to see if any discrepancies between the river polygons can be detected over the years. The overlay was performed consecutively from 2011 to 2018. The default snapping distance from QGIS is used. None of the overlay analyses produced any additional polygons (slivers). This result suggests that no significant water channel morphological changes

are observed during the eight years. The digitization and evaluation of high-resolution remote sensing data suggest that the river system in the Wading River wetland complex area has sustained little to no significant morphological changes over the eight years. Since the wetland setting and hydrologic structure and the associated complexity is the foundation for the land use land cover changes over time, the digitization exercise suggests that any changes in land use land cover in the WRC are likely not related to changes in the structure of the river system.

### 3.2. Coupling between In Situ Data and Remote Sensing Data: Supervised Classification

Using the training data set and the satellite images, we applied the random forest algorithm (RF), the support vector machine algorithm (SVM), and the neural network algorithm (NN) machine learning algorithms using the "raster" [92], "RStoolbox" [93], and "caret" [94] packages in R 4.2 [95]. This assessment does not include the summer of 2012 since only a panchromatic image was able to be obtained and did not provide the level of detail needed for this assessment. The average accuracy is calculated from the 5-fold cross-validation confusion matrix on each year (which are not reported to save space).

Cross-validation measures suggest that the random forest algorithm consistently produced satisfactory classification accuracy with an overall accuracy of approximately 94% (ranging from 84–97%) and Kappa index ranging from 0.80–0.96. Detailed classification accuracy of the three algorithms for each year is reported in Table 1. For illustration purposes, Figure 3 reports the classified images for the five land cover types through RF for the winter and summer times in 2018.

**Table 1.** Machine learning algorithms supervised classification accuracy reports for the summer and winter evaluations for 2011–2018.

| Year | RF Average Accuracy | RF Kappa | SVM Average Accuracy | SVM Kappa | NN Average Accuracy | NN Kappa |
|---|---|---|---|---|---|---|
| 2011 summer | 0.954 | 0.943 | 0.883 | 0.831 | 0.846 | 0.765 |
| 2011 winter | 0.933 | 0.916 | 0.902 | 0.858 | 0.865 | 0.804 |
| 2012 summer | NA | NA | NA | NA | NA | NA |
| 2012 winter | 0.94 | 0.925 | 0.902 | 0.860 | 0.873 | 0.819 |
| 2013 summer | 0.974 | 0.962 | 0.964 | 0.948 | 0.811 | 0.721 |
| 2013 winter | 0.838 | 0.798 | 0.673 | 0.506 | 0.703 | 0.586 |
| 2014 summer | 0.972 | 0.961 | 0.955 | 0.936 | 0.804 | 0.709 |
| 2014 winter | 0.943 | 0.918 | 0.929 | 0.899 | 0.843 | 0.773 |
| 2015 summer | 0.957 | 0.938 | 0.881 | 0.828 | 0.820 | 0.741 |
| 2015 winter | 0.941 | 0.926 | 0.878 | 0.824 | 0.731 | 0.612 |
| 2016 summer | 0.929 | 0.911 | 0.903 | 0.861 | 0.862 | 0.802 |
| 2016 winter | 0.946 | 0.932 | 0.934 | 0.906 | 0.875 | 0.821 |
| 2017 summer | 0.943 | 0.929 | 0.939 | 0.913 | 0.818 | 0.739 |
| 2017 winter | 0.967 | 0.959 | 0.939 | 0.913 | 0.833 | 0.757 |
| 2018 summer | 0.969 | 0.956 | 0.962 | 0.945 | 0.849 | 0.785 |
| 2018 winter | 0.901 | 0.876 | 0.859 | 0.798 | 0.737 | 0.617 |

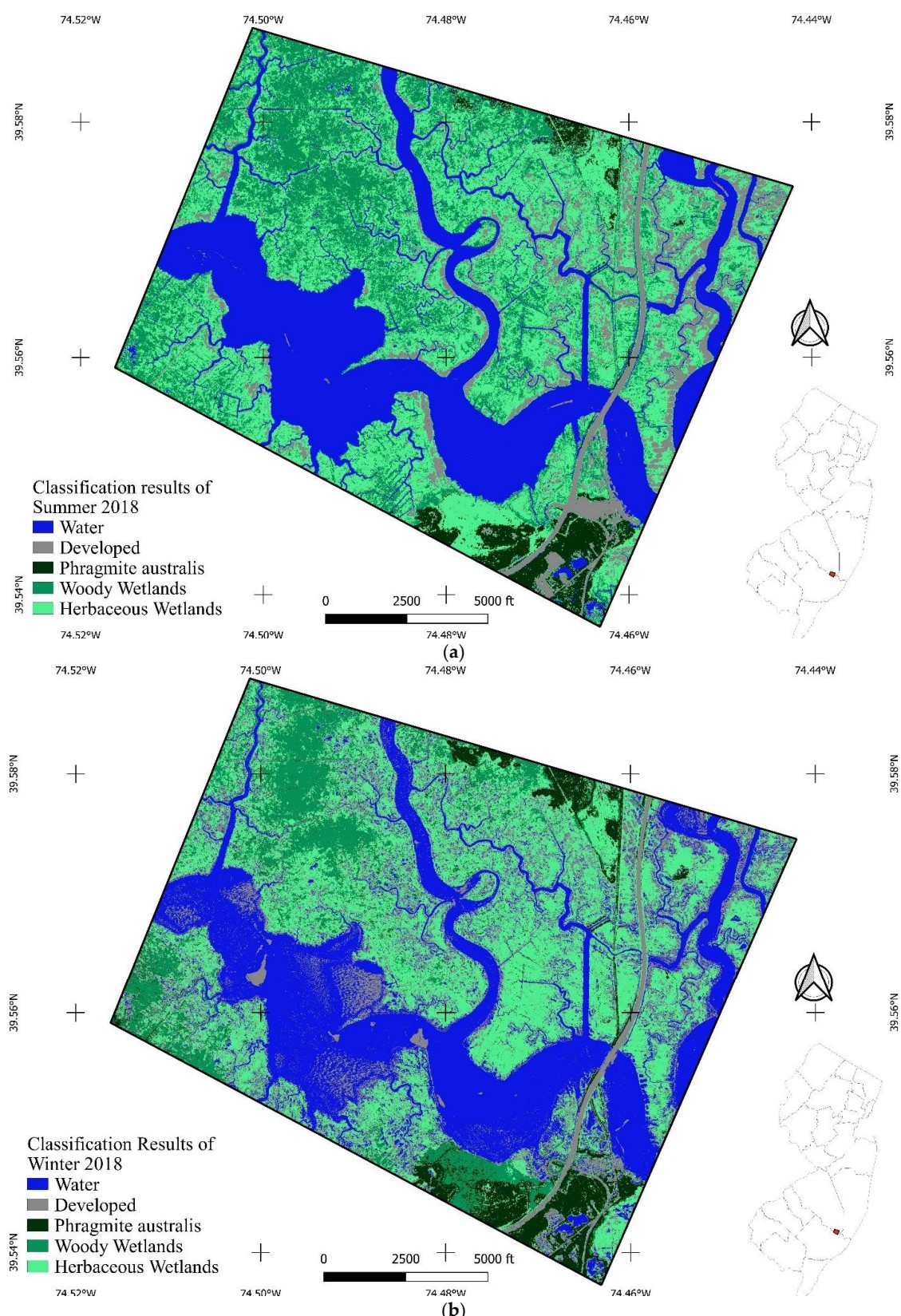

**Figure 3.** Random forest supervised classification of land covers in summer and winter 2018. (**a**) Classification results of summer 2018. (**b**) Classification results of winter 2018.

Although the accuracy statistics show the supervised classifications with random forest algorithm are acceptable, clear misclassification between high reflective water surface and developed land cover types was observable from the final classified images (Figure 3). In the summertime, some of the riverbanks were misclassified as developed lands. In the wintertime, the reflectance of the water/ice also made areas of water be reclassified as developed lands (Figure 3). Still, other than these obvious misclassifications, the overall image does express a reasonably accurate description of land use/land cover types in the WRC. The total areas based on the classified images for the five identified land cover types from 2011–2018 are summarized in Table 2 (summertime) and Table 3 (wintertime).

**Table 2.** Summertime areas of land cover types (square kilometers) based on random forest supervised classifications.

| Land Use | 2011 | 2013 | 2014 | 2015 | 2016 | 2017 | 2018 |
|---|---|---|---|---|---|---|---|
| Water | 5.10 | 5.50 | 5.13 | 5.14 | 5.34 | 5.56 | 5.86 |
| Developed | 1.28 | 1.07 | 1.88 | 2.54 | 1.80 | 1.99 | 2.27 |
| *Phragmites australis* | 1.47 | 1.63 | 0.80 | 1.53 | 1.02 | 0.86 | 0.80 |
| Woody Wetlands | 2.97 | 5.28 | 3.94 | 4.00 | 4.43 | 3.68 | 4.37 |
| Herbaceous Wetlands | 9.90 | 7.24 | 8.97 | 7.52 | 8.12 | 8.64 | 7.42 |

**Table 3.** Wintertime areas of land cover types (square kilometers) based on random forest supervised classifications.

| Land Use | 2011 | 2012 | 2013 | 2014 | 2015 | 2016 | 2017 | 2018 |
|---|---|---|---|---|---|---|---|---|
| Water | 5.25 | 4.95 | 4.11 | 5.23 | 4.57 | 5.26 | 5.30 | 6.32 |
| Developed | 1.66 | 1.20 | 3.63 | 1.48 | 2.01 | 0.94 | 0.82 | 4.09 |
| *Phragmites australis* | 2.01 | 1.48 | 1.62 | 1.37 | 1.54 | 2.04 | 1.31 | 1.31 |
| Woody Wetlands | 3.82 | 5.67 | 4.64 | 3.92 | 4.29 | 3.56 | 5.18 | 2.10 |
| Herbaceous Wetlands | 7.98 | 7.42 | 6.71 | 8.66 | 8.32 | 8.92 | 8.11 | 6.91 |

The immediately interesting pattern that can be observed from these two tables is that we noticed the fluctuation but seemingly decreasing trend of the invasive species *Phragmite australis*. However, there is a discrepancy between the wintertime and summertime. The result suggests a higher coverage of *Phragmite australis* during winter than summer. The trend, however, remains a decreasing one. Similarly decreasing land cover types also include the herbaceous wetland. Water area seems to fluctuate (more so in the wintertime than summertime), but such fluctuation is likely the result of misclassified water areas as developed land cover (Figure 3). Developed land remains relatively stable through the 8 years (the spike in the wintertime of 2018 is a result of misclassifying frozen water and creek banks as developed land, as can be seen in Figure 3), and so is woody wetlands. The Wading River Complex is a pristine wetland area, dominated by woody and herbaceous wetlands and water. The developed areas where human activities are present are largely restricted in the north and southeast corners of the evaluation area (a boatyard and campground in the southeast corner, and route 9 and garden state parkway on the east side of the study area). Visual inspection of the images suggests that the presence of *Phragmites australis* and human activities are closely related. Still, a distinctive trend is hardly present from observing the tables alone.

Moreover, from Tables 2 and 3, we can see a rather inconsistent classification of the developed land cover types, especially in the wintertime since developed land (road surface, rooftops, and other impervious surfaces) and frozen wetland and the edge of small creeks where water was likely frozen have a rather similar surface reflectance signatures (Figure 3b). As a matter of fact, comparing Tables 2 and 3, we see that the summertime provides more consistent classification results than the wintertime for different land use

types. This is to be expected as vegetation covers in wintertime tend to be more blurred with the icy conditions in the marsh wetland. While we only report the classification results for 2018, we have classified images for all years (except for summer 2012) for both winter and summer times. These resultant images are used to conduct land cover change analysis and summarize the change of each land cover type year by year. More importantly, via analyzing the temporal changes for the 8 years, with 15 temporal points, we attempt to cultivate covariations among different land cover types. The covariation information will be used to generate the system dynamic simulation, which will give a simulative future of how land use land covers might change over the next 25 years.

### 3.3. Building a Grey System Coupled System Dynamic Model to Evaluate the Change of Phragmites australis in the Wading River Complex

From observing the classified images of the Wading River Complex (Figure 3 only shows the two images for 2018 as an illustration, but we have classified images for all years from 2011–2018), we can see that the *Phragmites australis* are primarily distributed around developed lands where relatively intense human activities (i.e., road and buildings) are present, which agrees with the hypothesis that the invasiveness is a result from intensified anthropogenic activities. *Phragmites australis* might be introduced because of the clearing of this section of the wetland for road and boat yard construction. In other regions of the Wading River Complex where human activities are less observable, however, *Phragmites australis* appears in much lower coverage.

To build the covariation relationships among the different land cover types, we examined the changing trends of all five different land cover types from 2011–2018 in both summer and winter times very closely. In addition, as an illustration, pairwise scatterplots using both the summertime and wintertime data were produced to identify the potential covariations among different land cover types (Figure 4).

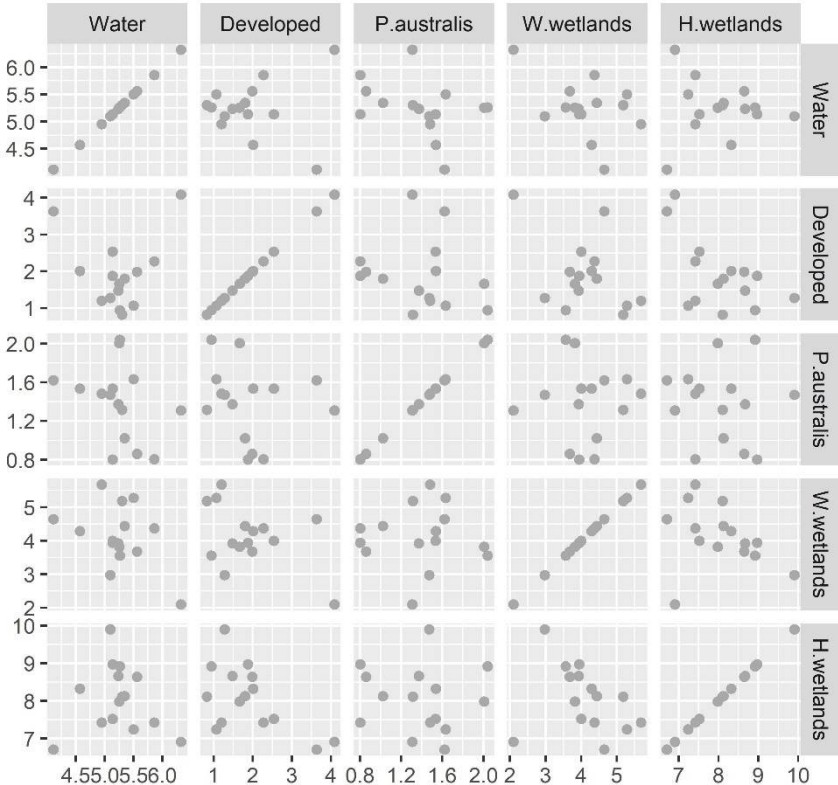

**Figure 4.** Pairwise scatterplots among the five different land cover types from 2011–2018 summer seasons (excluding 2012 summer, unit: square kilometers).

Examination of these outputs is the foundation for building a reasonably well-performing system dynamic simulative model since the future depends on the past and trends are seldom disrupted without significant disturbances. The pairwise scatterplots are simple tools to identify potential covariations among different variables so that dynamic relationships among different variables can be explored. The lack of longer time series data, however, suggests that any such exploration resulting from covariation must be treated with caution and should only be used as guidance to explore actual covariation among variables that will be used to construct the system dynamic model. If appropriate, the explored covariations will be compensated with grey-system-generated predictive series.

*3.4. Tentative Covariations among the Different Land Use Land Cover Types*

*Phragmites australis* is the focus of the simulation. To understand its invasiveness better, we designate this land cover type as the only stock variable in the model (Figure 2) and examine how other land cover types might impact the changing rate of *Phragmites australis* (the rate was not plotted in the pairwise plot). As a matter of fact, Figure 2 is also constructed after examining all the potential covariation patterns in Figure 4. By examining the *Phragmites australis* scatterplots (rows and columns, Figure 4), we can see that the changes over the years between *Phragmites australis* and developed, water, and herbaceous wetland land cover types show clear covariation. Although the short time series might prevent fully disclosing covariation from being established, the scatterplots provide a starting point to construct potentially meaningful covariation relationships. Since our goal is to identify the best possible covariations among different land cover types, we explored the relationships between the annual **changing rate** of *Phragmites australis* with all three land cover types. We explored linear and quadratic function forms between the changing rate of *Phragmites australis* and water, developed, and herbaceous wetlands. Using the fitted function's $R^2$ as an indicator, we found that a quadratic functional form (after removing the outliers in 2015) describes the relationships between the change rate of *Phragmites australis* land cover and the herbaceous wetlands produces the highest $R^2$:

$$\text{Annual change rate of } Phragmites = 6.0624 - 0.2049 \times \text{Herbaceous Wetlands} + 0.0016 \times \text{Herbaceous Wetlands}^2 \ (R^2 = 0.7598) \tag{6}$$

Then, after removing one outlier between the *Phragmites* and developed pairs in 2015 due to the low classification accuracy of developed land, we found the linear relationship between *Phragmites australis* and developed land covers produced the best $R^2$:

$$\text{Developed} = 19.023 - 1.2159 \times Phragmites \ (R^2 = 0.934) \tag{7}$$

Furthermore, after removing the irregular water data for 2013 and 2014, a negative covariation between *Phragmites* and water is also identified with the best $R^2$:

$$\text{Water} = 38.263 - 0.9254 \times Phragmites \ (R^2 = 0.8536) \tag{8}$$

In addition, there is a negative relationship between woody wetlands and herbaceous wetlands. The exploratory analysis suggests that a negative exponential relationship describes the changing trends well between the two land cover types that produced the best $R^2$:

$$\text{Woody wetlands} = 117.4 \times e - 0.029 \times \text{Herbaceous wetlands} \ (R^2 = 0.7248) \tag{9}$$

After removing an outlier in 2011, a quadratic relationship between herbaceous wetlands and developed land cover was also identified that produced the best $R^2$:

$$\text{Herbaceous wetlands} = 5.0001 + 8.4013 \times \text{Developed} - 0.3684 \times \text{Developed}^2 \ (R^2 = 0.693) \tag{10}$$

### 3.5. Further Refinement of the Tentative Covariation Relationships: Grey System Series Simulation

Although we explore the covariations among the components based on a balance between maximum fit (highest $R^2$) and least complexity (we usually prefer linear than quadratic relationships unless the change of $R^2$ is significant, and we never considered higher power polynomial relationships because of uncontrollable complexity might very well mask the true covariation than reveal it), all the above-identified relationships have somehow rigid linear or non-linear relationships. To ensure the simulations remain meaningful for a relatively longer term (25 years in the current design), we installed the Grey System Series Simulation model to control for unrealistic data generation and projection.

From Figure 2, we can see that the annual change rate of the *Phragmites australis* is at the center of the entire system since the amount of *Phragmites australis* distribution relates directly or indirectly to all other four types of land covers. Equation (1) suggests that the annual change of *Phragmites australis* cover can be best correlated with the change of herbaceous wetland in a quadratic form. *Phragmites australis* cover decreases as the herbaceous wetland cover increases to a point, then it starts to increase again. It must be emphasized here that there is no causality assumed. The observation is simply a quadratic covariation between the two land cover types. However, the quadratic form has the tendency to produce outputs that first become very small then very large. To prevent the quadratic equations to go beyond bounds, the grey system series simulative value for the annual change rate is added to ensure the change rate remains a reasonable amount. Equation (1) is then changed to:

$$\text{MIN } ((6.0624 - 0.2049 \times \text{Herbaceous} + 0.0016 \times \text{Herbaceous}^2 - 0.05638 \times (-1.08556)^{(\text{TIME} - 2)}) \tag{11}$$

The first part is the quadratic form from Equation (1), the second part, namely, $-0.05638 \times (-1.08556)^{(\text{TIME} - 2)}$ is the grey system simulative equation with $a = 0.004$, and $b = -0.195$. This grey system model has been implemented in the Grey System Prediction software package developed by Liu and colleagues [91]. As aforementioned, based on Liu and colleagues' experimental analysis, for the GM (1, 1) model, if the $a$ parameter that measures the inherent uncertainty of the target data series, is within the range of $[-0.3, 0.3]$, then the GM (1, 1) model can produce reasonably well projection for relatively longer term (up to 10–25 temporal periods, years in our study) [91]. Our model produced an $a$ that is well within the range of $[-0.3, 0.3]$, which suggests that for longer terms, the grey system series will take over the change and produce a more reasonable trend in the future.

Equations (2)–(5) remain unchanged since the relationships are not prone to produce drastically unrealistic results. In addition, a constraint that the sum of areas of all the different land cover types needs to stay constant (approximately 20 km$^2$) is added to prevent the simulation from producing unrealistic results. The model is then simulated in ISEE$^®$'s Stella Architect software to produce the most reliable simulation of how different types of land cover will evolve and change in the next decade to 25 years. We do not simulate beyond 25 years because the simulation will become increasingly unreliable due to the increasing possibility of unforeseeable factors in the long run [69,91]. The results produced from the simulation are reported in Table 4.

The simulative practice brings some very interesting and thought-provoking results which we hope could serve as an important incubator for thoughts and ideas of sustainable wetland restoration and management. First, one immediate impression from Table 4 is that developed land cover will continue to decrease, along with herbaceous wetland and water. Developed land in the study area is primarily composed of roads (Route 9 and garden state parkway) in the east, and a boatyard and campground in the southeast. The road is not likely to be extended or new roads built in the next 25 years. The boatyard and campground are seasonally used man-made structures and usually see the highest traffic during summertime. We expect the usage will remain rather stable or even less so to protect the pristine wetland ecosystem as governments' efforts for and residents' understanding of sustainable coastal land preservation and development broaden. On the other hand, the invasive *Phragmites australis* and the woody wetlands will increase consistently. Although

the area after around five years (2023) will become less of a prediction but more of an indication of future trends, it is still alarming to see expected increases in *Phragmites australis* in the near future. While we do observe a negative covariation between developed land and *Phragmites australis* in our initial exploration (Equation (2) and Figure 4), the decrease of developed land in the WRC did result in a decrease of *Phragmites australis*. We contend that while the biological invasion followed its introduction, its further expansion does not necessarily rely on intensified anthropogenic activities. The simulative model suggests that once the biological invasion took hold in an ecosystem, it could self-sustain and expand to compete with native species, regardless of whether anthropogenic activities intensify or not. This simulated result is particularly alarming as many anthropogenic activities are usually regarded as one of the decisive factors that introduce biological invasion and reduce biodiversity within an ecosystem [10,96–102]. While the simulative result does not suggest that anthropogenic activities are not related to increased biological invasion, we contend that devising an effective invasive species management plan requires detailed and long-term monitoring of the **covariations** instead of assuming a hypothetical causal relationship between two factors. This is because the full structure and information of a complex and multi-component system is usually a grey box, and we could only observe part of the system. Only through constant monitoring and long-term data collection, will an effective management plan be successful.

**Table 4.** Results of the Grey system coupled system dynamic simulation of land cover changes from 2019–2033. Units are in square kilometers.

| Year | Water | *Phragmites* | Developed | Herbaceous Wetland | Woody Wetland |
|------|-------|--------------|-----------|--------------------|---------------|
| 2019 | 4.76 | 1.47 | 1.25 | 7.71 | 4.64 |
| 2020 | 4.69 | 1.55 | 1.16 | 7.45 | 4.87 |
| 2021 | 4.68 | 1.56 | 1.14 | 7.40 | 4.92 |
| 2022 | 4.66 | 1.58 | 1.12 | 7.34 | 4.97 |
| 2023 | 4.62 | 1.63 | 1.06 | 7.13 | 5.16 |
| 2024 | 4.51 | 1.75 | 0.92 | 6.59 | 5.69 |
| 2025 | 4.38 | 1.88 | 0.76 | 5.83 | 6.53 |
| 2026 | 4.23 | 2.04 | 0.56 | 4.79 | 7.89 |
| 2027 | 4.22 | 2.06 | 0.54 | 4.69 | 8.03 |
| 2028 | 4.21 | 2.07 | 0.53 | 4.59 | 8.17 |
| 2029 | 4.20 | 2.08 | 0.51 | 4.50 | 8.32 |
| 2030 | 4.18 | 2.10 | 0.50 | 4.40 | 8.47 |
| 2031 | 4.17 | 2.11 | 0.48 | 4.29 | 8.63 |
| 2032 | 4.16 | 2.12 | 0.46 | 4.19 | 8.79 |
| 2033 | 4.14 | 2.14 | 0.44 | 4.08 | 8.96 |

Additionally, the inferred trend produced by this 25-year simulative assessment is very informative. From the classification practices, we did not have sufficient evidence to suggest a clear trend of the expansion of *Phragmite australis*. Through simulation, albeit a simple one, we see a predicted increase in *Phragmites australis* and woody wetlands, which practically suggests that this small tidal marsh area could be under stress. While studies often suggest that *Phragmite australis* do not necessarily have a significant impact on nekton species or even provide short-term ecoservices [103–105], it is also found that the removal of *Phragmite australis* provided enhanced conditions for nekton species [103]. With more *Phragmite australis* and woody wetlands, there is a potential that this tidal marsh could reduce biodiversity in the long term. It is important to note that this analysis is not able to directly answer which land cover type is replacing which, though with only five land cover types in this simple system, the increasing of *Phragmite australis* and woody wetlands is at the expense of decreased water and herbaceous wetlands. Still, as aforementioned, the existence and gradual spreading of *Phragmites australis* could pose a threat to local fauna

and flora. This simulation model suggests that based on recent conditions, it is likely that *Phragmites australis* cover can and will increase in the near future. The magnitude of such an increase is speculative though. While simulative model results should not be treated as actual predictions, the trends revealed by the simulative model, however, shall provide guidance for understanding the baseline conditions of the pristine wetland complex in the WRC and similar areas with a well-rooted *Phragmites australis* population.

## 4. Conclusions

Employing the random forest machine learning algorithm, we classified eight years of high-resolution remote sensing images of the WRC into five dominant land cover types. In so doing, this research is able to successfully identify the change in the coverage of the invasive species *Phragmite australis* in this relatively pristine tidal marsh wetland from 2011–2018. While a clear trend of the invasive species is not readily present during the study period, we developed a grey system coupled system dynamic model via exploring the covariation among the five land cover types to simulate how these land covers will likely change in the near future. While the available information is limited for the simulation, the simulative results are still indicative that the invasive species *Phragmites australis* is likely to increase in the WRC. Regardless of the origin of the population, the existence and continuous spreading of *Phragmites australis* could pose a threat to local fauna and flora. Lacking traces of the invasive pathways of *Phragmites australis* in the Wading River Complex prevents any meaningful causality relationship between the coverage of *Phragmites australis* and other factors from being established. Yet through utilizing the covariation patterns among the land cover types, we are able to establish a simple but useful model that could provide a dynamic scenario for how land use land cover in WRC likely changes in the future, and targeted management and restoration efforts could be devised.

The outcome of this work provides important perspectives in the context of wetland restoration, monitoring, and management. Components of this study directly complement the goals and vision outlined in the EPA-approved Wetland Program Plan in New Jersey (NJWPP) [106] and provide valuable information in a minimally disturbed wetland complex for a wide range of stakeholders involved in ecological restoration and natural resource conservation.

A potential future extension of the current study, which could be extremely beneficial for local wetland management and nekton species inventory, would be to connect surveys of secondary production with remotely sensed landscape variables to establish an index that can accurately reflect potential changes in the relative abundance of local secondary nekton species. Recent studies that include monitoring water quality, including nutrients, and soil salinity through remote sensing techniques [107,108] could be a potential direction to be pursued in future studies. In addition, the increasing availability of open access remote sensing data, such as the ones available from PlanetScope, provides limitless opportunities to further extend the current research for verification of the system dynamic model and fine-tuning. This will be our immediate next step in the chain of research initiated from this study.

**Author Contributions:** Conceptualization, D.Y. and N.A.P.; methodology, D.Y. and C.F.; software, D.Y. and C.F.; validation, D.Y. and N.A.P.; formal analysis, D.Y.; investigation, D.Y.; resources, N.A.P.; data curation, D.Y.; writing—original draft preparation, D.Y. and N.A.P.; writing—review and editing, D.Y. and N.A.P.; visualization, D.Y.; supervision, D.Y. and N.A.P.; project administration, N.A.P.; funding acquisition, D.Y. and N.A.P. All authors have read and agreed to the published version of the manuscript.

**Funding:** This study is supported in part by the EPA funded grant # CD96275900.

**Data Availability Statement:** Not applicable.

**Conflicts of Interest:** The authors declare no conflict of interest.

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
