# Peer review of "Simulating the Changes of Invasive Phragmites australis in a Pristine Wetland Complex with a Grey System Coupled System Dynamic Model: A Remote Sensing Practice"

_remotesensing, doi:10.3390/rs14163886_

Round 1
Reviewer 1 Report
Dear Authors,
Thank you for this interesting study. Saving time and improving accuracy and precision is some of the basic reasons for the application of remote sensing based systems to scientific studies, especially to the environmental researches. However, your application of vector based data to this study (digitizing of the study area) is controversial to this idea. It is obvious that both methods (digitizing and automated classification) have advantages of their own (e.g. lacking of bias, being analytic). I believe, it would be better to use only the classified images as long as the methods are consistent. Could you please explain why did you use method more explicitly.
Author Response
Response: Thank you very much for your insightful comments. We agree that you are making a valid argument here regarding the vector and raster-based analyses.
We provided a digitization analysis to extract the morphological information of the river channels over the years. We did that in case that the dynamics of the river channels might also contribute to the change of land use and land cover in the wetland complex. If the morphology of the river channels does change quite significantly, it will be added as a factor in later system dynamics simulative models. The digitization practice, however, suggests that the morphology of the river channels remained unchanged over the period of our study (2011 – 2018), which allows us to focus our system dynamic simulative model with only classified the land use land cover information. We added a brief explanation in the text stating just that (page 5, lines 209 – 211). Thank you again for this comment; it makes the manuscript more focused now. The digitization analysis sub-section in the results and discussion section serves as a summary of this brief finding.
Reviewer 2 Report
Review of "Simulating the changes of invasive Phragmites australis in a pristine wetland complex with a grey system coupled system dynamic model: a remote sensing practice."
By
Danlin Yu, Nicholas A. Procopio and Chuanglin Fang
In general it is a very interesting paper with a creative approach to environmental forecasting. As far as i see, the numeric methods are well implemented but I have some observations and suggestions that i'm sure will enhance the applicability and benefit of this research's results.
Figure 1 and all the rest of the maps require geographical references (a geographical coordinates grid -standard for publications-, and besides not everybody know where New Jersey is).
lines 423-424, A useful data exploration, to contextualize this variability, would be to obtain the records of average rainfall per season, which would probably allow the authors to explain the variability in the water class area, or to identify a source of error in the predictions of the model.
Also a DTM incorporated into the GIS, could point out the correspondence between the variable water class areas and the lowest elevations, thus sorting out the missclassifications with the expected change in low areas.
Lines 444-445, When the authors state that the water area did not have a significant variability in the study period, and having digitized the whole water body, I wonder why they didn't just create a mask for water, and use it as a constant (which it is, since according to the analysis there is no significant change over time), that would sort out the misclassification of riverbanks and developed area, and other water misclassifications that could be entering additional error to the predictive model.
Lines 448, 452, 468 Authors have made clear that issue before, probably they just need to keep one of these statements regarding the missing season, please pick the most relevant.
Lines 468-472 Is there a discernable effect of previous land clearing towards the increase of P. australis? Meaning, that you might have had the clearing of an area for building something -like a road- and then that "excess clearing" -like access roads and machinery yards- after the construction is finished, is colonized by P. australis. Like and early colonizer that dominates for a while and then gives way to other vegetation types -like a succesional species assemblage-? Which is more or less what you describe on lines 566-568.
table 4. The increasing trend in P. australis and the decreasing one in Developed area, clash with the trends you described earlier. One could assume the missclassifications of water/developed area in the different times had a role in this prediction. So This result must be taken with a grain of salt, as you later recognize on the text.
Since this table is the main result of this modelling excercise, and it has some counterintuitive results, what i suggest now, is to contrast the results of the model over the classification of imagery from 2019-2022. Even if there are no more funds to acquire additional WV-2 imagery - which i fully understand, having been there myself- there are other high-resolution imagery sources (like planet.com) that have academic licenses agreements. Authors can compare their WV-2 2018 summer and winter classifications with classifications of Planetscope imagery (most probably there are scenes from the exact same dates of thier WV-2), to calibrate between imagery types, and then compare the resulting classifications of the following years with their model's predictions.
This way, the authors could actually sustain the statement about providing accurate scientific data for the management decision support. Specially because this is a theoretical numerical approach (which, for its implementation i don't see any problem) and not a causal modeling, as they also state in the manuscript.
In the one hand the authors will have a great opportunity to validate their model with real data, or in the other hand, they will have the chance to calibrate their model with data from a longer time series.
Author Response
Review of "Simulating the changes of invasive Phragmites australis in a pristine wetland complex with a grey system coupled system dynamic model: a remote sensing practice."
By
Danlin Yu, Nicholas A. Procopio and Chuanglin Fang
In general it is a very interesting paper with a creative approach to environmental forecasting. As far as i see, the numeric methods are well implemented but I have some observations and suggestions that i'm sure will enhance the applicability and benefit of this research's results.
Response: Thank you very much for your comments and insightful reviews. We are grateful for your comments and will strive to fully address them.
Figure 1 and all the rest of the maps require geographical references (a geographical coordinates grid -standard for publications-, and besides not everybody know where New Jersey is).
Response: Geographic reference with longitudes and latitudes is added for all the maps. Thank you for this important cartographic design point.
lines 423-424, A useful data exploration, to contextualize this variability, would be to obtain the records of average rainfall per season, which would probably allow the authors to explain the variability in the water class area, or to identify a source of error in the predictions of the model.
Also a DTM incorporated into the GIS, could point out the correspondence between the variable water class areas and the lowest elevations, thus sorting out the missclassifications with the expected change in low areas.
Response: These are certainly very important points to consider. We did not add these types of information in our current study because of two reasons: First, the study area is small (20 km2), variations of the precipitation and elevation (especially elevation) might not provide enough explaining power to account for the mis-classification problem. Second, the misclassification, as we have experimented for quite some time during training site creation, is mainly because of the reflectivity of the water when the images were taken. We have tried to create the training dataset that contains varied water samples but found the training data provided worse classification results than if we only include relatively homogeneous samples in the training dataset. But this is a very important point, and we added a brief explanation in the text to clarify that.
Lines 444-445, When the authors state that the water area did not have a significant variability in the study period, and having digitized the whole water body, I wonder why they didn't just create a mask for water, and use it as a constant (which it is, since according to the analysis there is no significant change over time), that would sort out the misclassification of riverbanks and developed area, and other water misclassifications that could be entering additional error to the predictive model.
Response: Thank you for pointing this issue out. When we performed the digitization, we did intend to use the vector layer (polygon) to calculate the water area, but after a few rounds of discussion among the authors, we realized that such practice might not serve our purpose well especially in the small study area (20 km2) since the area calculation could be impacted by accidental digitization errors on raster images. We eventually decided to use the digitization of the river channels for morphological changes detection only. We have added this clarification in the revision and hope the clarification could gain your approval.
Lines 448, 452, 468 Authors have made clear that issue before, probably they just need to keep one of these statements regarding the missing season, please pick the most relevant.
Response: thank you for spotting this repetition. We decide to keep the first statement when it appears and remove the rest.
Lines 468-472 Is there a discernable effect of previous land clearing towards the increase of P. australis? Meaning, that you might have had the clearing of an area for building something -like a road- and then that "excess clearing" -like access roads and machinery yards- after the construction is finished, is colonized by P. australis. Like and early colonizer that dominates for a while and then gives way to other vegetation types -like a succesional species assemblage-? Which is more or less what you describe on lines 566-568.
Response: the authors discussed this point during the revision, and we agreed that without further information and data, we cannot be certain that the previous land clearing had a clearly discernable effect on the increase of P. australis. But we do agree that the probability exists, and we added this speculation in the text to reflect such probability.
table 4. The increasing trend in P. australis and the decreasing one in Developed area, clash with the trends you described earlier. One could assume the missclassifications of water/developed area in the different times had a role in this prediction. So This result must be taken with a grain of salt, as you later recognize on the text.
Response: Thank you for your comments. This is indeed one of the points that we intend to make when treating classified images and using the information for simulative modeling. We emphasized this point in this round of revision to make sure that our purpose for the current study is to present a viable way to combine remote sensing image information and simulative modeling for wetland biological invasion management practices. The trend and system dynamics are the foci in this management practice, the actual numbers are to be taken with caution.
Since this table is the main result of this modelling excercise, and it has some counterintuitive results, what i suggest now, is to contrast the results of the model over the classification of imagery from 2019-2022. Even if there are no more funds to acquire additional WV-2 imagery - which i fully understand, having been there myself- there are other high-resolution imagery sources (like planet.com) that have academic licenses agreements. Authors can compare their WV-2 2018 summer and winter classifications with classifications of Planetscope imagery (most probably there are scenes from the exact same dates of thier WV-2), to calibrate between imagery types, and then compare the resulting classifications of the following years with their model's predictions.
This way, the authors could actually sustain the statement about providing accurate scientific data for the management decision support. Specially because this is a theoretical numerical approach (which, for its implementation i don't see any problem) and not a causal modeling, as they also state in the manuscript.
In the one hand the authors will have a great opportunity to validate their model with real data, or in the other hand, they will have the chance to calibrate their model with data from a longer time series.
Response: Thank you for your insightful suggestion. This suggestion is very enticing, and we immediately went to PlanetScope’s website excitedly attempting to find relevant images intending to follow your suggestion. What dismayed us is that the number of available data is simply overwhelming, and to find the images that contain the exact place and time as in our current study could be very time-consuming. Still, we tried the Copernicus Open Access Hub portal and used an area tool to extract Sentinel-3 images for our study area. We ended up giving up extracting the data giving the time constraint we have to revise the manuscript. But the suggestion is certainly of great value, and we added a future research direction at the end of our manuscript indicating exactly that we will follow through the suggestion and conduct more robust analysis with these open access datasets. We hope this somehow failed but hopefully understandable attempt could gain your approval.
Reviewer 3 Report
The manuscript with the title “Simulating the changes of invasive Phragmites australis in a pristine wetland complex with a grey system coupled system dynamic model: a remote sensing practice” is an attempt to use remote-sensed data for mapping invasive Phragmites australis and use the outcome to simulate land cover change for the next 25 years. While the idea and the approach of authors look promising, the current version of the manuscript contains many critical problems that sadly I must reject:
- First, their paper structure is questionable. They tried to explain their methodology in result and discussion section (L384-391, L459-464, L494-510, L549-562). It is hard to read the manuscript logically and I recommend authors to reconstruct their manuscript first
- Second, this manuscript contains qualitative claims without information or evidence to support them. For example, L370-379 shouldn’t appear in a scientific paper because reviewers or audiences cannot verify this claim (no images, no info, no link to support that)
- Third, their classification results are unreliable. The main problem is the accuracy of land-classification. From the look of figure 3, the average accuracy is around than 80% or less. Another example is that water area was 6.32 square kilometres in the 2018 winter, higher than the summer one of 2018 (5.86 km2) though the result of misclassified water the result of misclassified water areas as developed areas. Somehow, I doubt that their sampling step was not good enough and there is no information of how they divided their samples to train, validate, and test set. Authors need to provide information of areas (Ex: figure(s)) they generated/created to extract their sample data.
- Moreover, authors decided to choose random forest algorithm with words only. In my opinion, they need to provide results of other algorithms rather than putting them in the methodology section and no results after that. Confusion matrices are needed to see the reason of inaccurate result as well.
- Some other minor problems:
o Reference in this manuscript should follow the format of RS journal (Ex: square brackets instead of round brackets for references)
o Some text errors/mistakes (Ex: L49 – “about$1100”)
o L198: please provide the wavelength of bands
o L214-215 (Figure 1): Please provide the timing of the image, longitude and latitude information of the image as well.
Author Response
The manuscript with the title “Simulating the changes of invasive Phragmites australis in a pristine wetland complex with a grey system coupled system dynamic model: a remote sensing practice” is an attempt to use remote-sensed data for mapping invasive Phragmites australis and use the outcome to simulate land cover change for the next 25 years. While the idea and the approach of authors look promising, the current version of the manuscript contains many critical problems that sadly I must reject:
Response: Thank you for your critical review. We are hoping that our revision with your insightful comments as guidance could gain your approval for final acceptance.
- First, their paper structure is questionable. They tried to explain their methodology in result and discussion section (L384-391, L459-464, L494-510, L549-562). It is hard to read the manuscript logically and I recommend authors to reconstruct their manuscript first
Response: We carefully restructured the manuscript to avoid repeating our methodology in the result and discussion sections as detailed in your instructional comments. L384-391, L459-464, and L549-562 were now moved to the methodological section with slight modification. L494-510 (and all the following resulting covariation equations), however, describe the results from the covariation mining following the image classification and pairwise correlation analysis. We retained this section in the results and hope the revision will gain your approval.
- Second, this manuscript contains qualitative claims without information or evidence to support them. For example, L370-379 shouldn’t appear in a scientific paper because reviewers or audiences cannot verify this claim (no images, no info, no link to support that)
Response: Our initial analysis was done based on visual comparison. We performed map overlay analyses using the digitized polygons from 2011 to 2018 consecutively to verifying if there are any significant discrepancies between the polygons over two consecutive years. None of the overlay produced any detectable discrepancies using the default snapping distance in QGIS. We added this result to the discussion and hopefully this addition will gain your approval.
- Third, their classification results are unreliable. The main problem is the accuracy of land-classification. From the look of figure 3, the average accuracy is around than 80% or less. Another example is that water area was 6.32 square kilometres in the 2018 winter, higher than the summer one of 2018 (5.86 km2) though the result of misclassified water the result of misclassified water areas as developed areas. Somehow, I doubt that their sampling step was not good enough and there is no information of how they divided their samples to train, validate, and test set. Authors need to provide information of areas (Ex: figure(s)) they generated/created to extract their sample data.
Response: For 2018 summer, we have a 0.956 Kappa coefficient, and for winter, we have a 0.876 Kappa coefficient (Table 1). The accuracy of classification in the winter is indeed low because of misclassification of some of the water areas as developed area, and some of the woody and herbaceous wetlands are misclassified as water area. The overall accuracy is still acceptable, but the selection of the training sites (which also serve as validation sites for supervised classification) indeed has an impact on the classification results because creating land use land cover spectral signature is heavily influenced by pixels included in the training sites. As a matter of fact, we have experimented for quite some time during training site creation because of the reflectivity of the water when the images were taken. We have tried to create the training dataset that contains varied water sample pixels but found the training data provided worse classification results than if we only include relatively homogeneous sample pixels in the training dataset. But this is a very important point, and we added a brief explanation in the text to clarify that. We also updated Figure 1 to include the locations of the training sites for various land cover types.
- Moreover, authors decided to choose random forest algorithm with words only. In my opinion, they need to provide results of other algorithms rather than putting them in the methodology section and no results after that. Confusion matrices are needed to see the reason of inaccurate result as well.
Response: Thank you for your suggestion regarding the efficiency of the algorithms. We do agree that it will be a standard practice to report the accuracies of different machine learning algorithms for comparison purposes and reporting the confusion matrix to support the accuracy claim. We do take your advice and added the average accuracy and Kappa coefficients for the other two machine learning algorithms (support vector machine and neural network) to table 1 to show the better performance of the random forest algorithm. We hope this addition will gain your approval.
For the confusion matrices, in the current study, we have eight years and 15 images of classification. Reporting 15 confusion matrices will be redundant and might not add additional contribution to the ensuing discussion, especially since classification is not the only theme of the current research. We hence only report the average accuracy and Kappa coefficients for clarity. Our purpose is to extract relatively accurate land use land cover information and use the information to build a system dynamic model for simulation and biological invasion management.
- Some other minor problems:
o Reference in this manuscript should follow the format of RS journal (Ex: square brackets instead of round brackets for references)
Response: Thank you for your meticulous review. We used Endnote reference management system to change the reference to agree with the journal’s requirements.
o Some text errors/mistakes (Ex: L49 – “about$1100”)
Response: Thank you for the detailed review. We have corrected that by add the required space.
o L198: please provide the wavelength of bands
Response: Wavelengths of the bands are added.
o L214-215 (Figure 1): Please provide the timing of the image, longitude and latitude information of the image as well.
Response: The maps are all re-created with timing and geographic reference information added.
Round 2
Reviewer 3 Report
I agree with authors' replies and the revision of the manuscript. I believe the manuscript is ready for publication.